# Functionalized *Moringa oleifera* Gum as pH-Responsive Nanogel for Doxorubicin Delivery: Synthesis, Kinetic Modelling and In Vitro Cytotoxicity Study

**DOI:** 10.3390/polym14214697

**Published:** 2022-11-03

**Authors:** Sunita Ranote, Marta Musioł, Marek Kowalczuk, Veena Joshi, Ghanshyam S. Chauhan, Rakesh Kumar, Sandeep Chauhan, Kiran Kumar

**Affiliations:** 1Centre of Polymer and Carbon Materials, Polish Academy of Sciences, 34. M. Curie-Skłodowska St., 41-819 Zabrze, Poland; 2Department of Chemistry, Hemvati Nandan Bahuguna Garhwal University, SRT Campus, Tehri Garhwal, Srinagar 249199, Uttarakhand, India; 3Department of Chemistry, Himachal Pradesh University, Summer Hill, Shimla 171005, Himachal Pradesh, India

**Keywords:** *Moringa oleifera* gum nanogel, doxorubicin delivery, pH-responsive, release kinetics, *Rhabdosarcoma* cells

## Abstract

Environment-responsive-cum-site-specific delivery of therapeutic drugs into tumor cells is a foremost challenge for chemotherapy. In the present work, *Moringa oleifera* gum–based pH-responsive nanogel (MOGN) was functionalized as a doxorubicin (DOX) carrier. It was synthesized via free radical polymerization through the γ-irradiation method using acrylamide and N,N’-MBA followed by hydrolysis, sonication, and ultracentrifugation. The swelling behavior of MOGN as a function of pH was assessed using a gravimetric method that revealed its superabsorbent nature (365.0 g/g). Furthermore, MOGN showed a very high loading efficiency (98.35 %L) of DOX by MOGN. In vitro release studies revealed that DOX release from DOX-loaded MOGN was 91.92% at pH 5.5 and 12.18% at 7.4 pH, thus favorable to the tumor environment. The drug release from nanogel followed Korsmeyer–Peppas model at pH 5.5 and 6.8 and the Higuchi model at pH 7.4. Later, the efficient DOX release at the tumor site was also investigated by cytotoxicity study using *Rhabdomyosarcoma* cells. Thus, the synthesized nanogel having high drug loading capacity and excellent pH-triggered disintegration and DOX release performance in a simulated tumor environment could be a promising candidate drug delivery system for the targeted and controlled release of anticancer drugs.

## 1. Introduction

With the advent of newer techniques and treatment methods in the medical world, exponential advancement has been made in cancer biology. Despite the momentous progress in the field of cancer treatment, it is still the foremost cause of death, causing 9.9 million deaths in 2020, out of which around 70% of deaths were reported from developing countries and has proved to be the most noticeable barrier to the long-life expectancy of the human race [1,2,3]. Several chemotherapeutic drugs have been developed to combat cancer. But their efficacy is limited as most of these drugs have been associated with major shortcomings, viz. poor water solubility and the absence of specificity for cancerous tissues resulting in poor antitumor efficiency, toxicity to the normal tissues, and several side effects such as cardiomyopathy, nephrotoxicity, neurotoxicity, bone marrow suppression and drug resistance [4,5,6]. Most of these drugs are administered intravenously via injections which results in initial burst release and leads to their consecutive decay; hence, their concentration reaches below the therapeutic level in the blood. Thus, drug specificity is the foremost challenge in the medical field, and it can be overcome by developing efficacious drug-delivery systems with high therapeutic specificity.

Several drug delivery systems have been designed and reported as potential carriers for anticancer drugs, viz. polymeric micelles [7], polypeptides [8], polymeric nanoparticles [9,10,11,12], gold nanoparticles [13,14], liposomes [15], silica [16,17], polymersomes [18], nanocomposites [19], nanocapsule [20] and nanogels [21,22,23,24,25,26,27,28,29,30,31,32]. Among these, stimuli-responsive nanoscale drug delivery systems have attracted researchers’ attention worldwide as pH-responsive drug nanocarriers. They have the potential to carry a large amount of drug due to their very high surface/volume ratio and their ability to selectively release the encapsulated drug molecules to the acidic tumor microenvironment either into the lysosomes/late endosomes or escape therefrom to the cytoplasm of the cancer cell [33,34].

For the past few years, a broad range of pH-responsive polysaccharide-based nanocarriers have become the promising subject of research for site-specific anticancer drug delivery as they are cost-effective, biodegradable, biocompatible, non–toxic, and show better compliance with the patient body [26,27,28,31,35,36,37]. Moreover, among several nanocarriers, polysaccharide-based nanogels have emerged as the most suitable vehicles for the delivery of anticancer drugs to the targeted site as they possess unique characteristics of large surface area, ease of modification to multifunctional materials, high water adsorption capacity, low interfacial tension with biological fluids and tissues, soft and smooth surface which minimize frictional irritation to the normal tissues [24,34,37,38]. Various studies on designing polysaccharide-based nanogels from various materials, viz. carboxymethyl chitosan, dextrin, chitin, hyaluronic acid, alginate, cellulose, pullulan, and xanthan gum, among others, have been reported in the literature. These nanogels successfully delivered an anthracycline anticancer drug, doxorubicin [22,23,24,25,26,27,29,30,31,32,33,39,40].

In view of the above-discussed subject matter, a new nanogel was synthesized from plant polysaccharide, *Moringa oleifera (M. oleifera)* gum (MOG) using acrylamide (AAm) and N,N’-methylenebisacrylamide (N,N’-MBA) via γ-irradiation technique followed by partial hydrolysis, sonication, and ultracentrifugation. Later, the synthesized nanogel was evaluated as a carrier for the anticancer drug doxorubicin (DOX). In the present work, keeping in view of the acidic environment of the tumor tissue (pH 6.8 and more acidic in intracellular compartments such as endosomes and lysosomes (5.0–6.5)), acid-sensitive and labile carboxyl groups were introduced via partial hydrolysis of the amide groups of poly(AAm) on *M. oleifera* gum nanogel (MOGN) surface, and these electrostatically interact with the protonated amino groups (–NH_3_^+^) of DOX [25] to ensure maximum loading and simultaneously ensuring the maximum release of the DOX by its pH-induced swelling behavior. Synthesized nanogel has a very high-water absorption capacity that enables the partitioning of the drug from its solution to the nanogel and results in a high drug–loading capacity. The latter, combined with the pH-responsive nature of the nanogel, renders the reported nanogel an effective carrier and release device of DOX at the tumor site, pH 5.5. In addition, in vitro cellular cytotoxicity was also investigated using RD (*Rhabdomyosarcoma*) cells as rhabdomyosarcoma (RMS), a most common soft tissue sarcoma (STS), accounts for over 50% of all the STS detected in teens and children [41]. Moreover, DOX acts as a substrate for ABC transporters present in RD cells, and hence, these cells accrue a substantial amount of DOX chiefly in the lysosomes and nucleus [42]. Thus, the present work highlights the utilization of plant gum, *Moringa oleifera* gum, as an efficient and promising candidate for the targeted delivery of the anticancer drug DOX. To the best of our knowledge, no work related to the synthesis of superabsorbent *M. oleifera* gum nanogel and its application as a pH-responsive delivery of DOX has been reported.

## 2. Materials and Methods

### 2.1. Reagents and Materials

*M. oleifera* gum (MOG) was collected from Srinagar, Uttarakhand, India. Doxorubicin hydrochloride (DOX, Samarth Life Sciences Pvt. Ltd., Mumbai, Maharashtra, India), disodium hydrogen orthophosphate anhydrous (Na_2_HPO_4_), sodium dihydrogen orthophosphate dihydrate (NaH_2_PO_4_.2H_2_O), N,N’-bismethyleneacrylamide (N,N’-MBA), acrylamide (AAm), methanol, hydrochloric acid (HCl), sodium hydroxide (NaOH), dimethyl sulfoxide (DMSO) (S.D Fine–Chem Ltd., Mumbai, Maharashtra, India), Dulbecco’s Modified Eagle’s Medium (DMEM), 3-(4,5-dimethylthiazol-2-yl)-2,5-diphenyltetrazolium bromide (MTT), pyridoxal 5’-phosphate (PLP), fetal bovine serum (FBS), penicillin and streptomycin (HiMedia Lab. Pvt. Ltd., Mumbai, Maharashtra, India) and *Rhabdomyosarcoma* cells (RD cells, National Culture for Cell Science (NCCS), Pune, Maharashtra, India), were of analytical grade. These were used as received. Double distilled water was used in all the experiments.

### 2.2. Preparation of Moringa oleifera Gum Nanogel (MOGN) and DOX-Loaded MOGN (DOX–MOGN)

Purification of crude MOG was performed as per our previously reported work [43]. MOG hydrogel was synthesized via the γ–irradiation method [44]. Briefly, AAm and MOG solution were mixed in a 1:1 weight ratio with constant stirring and then left the mixture undisturbed for 6 h. To the mixture, we added 1% N,N’-MBA (crosslinker) by weight of the mixture components, and it was irradiated in a gamma chamber (^60^Co–rays) at a dose rate of 0.6 KGy/h for 24 h with a total γ–irradiation dose of 14.4 KGy. After irradiation, the synthesized MOG–hydrogel (MOGH) was equilibrated with distilled water for 72 h to remove any unreacted fractions. Thereafter, MOGH was dried at 50 °C in an oven. To 2.5% solution of MOGH was added 50.0 mL of 2.5 M NaOH to affect its partial hydrolysis, and the contents were stirred for 5 h at 40 °C. Thereafter, 1 N HCl was added under stirring and that was followed by adding acetone for precipitation of the hydrolyzed hydrogel. Finally, the precipitates obtained were washed with 30% methanolic solution and dried at 50 °C in an oven [45]. Later, the hydrolyzed MOGH was converted to nanoform, *Moringa oleifera* gum nanogel (MOGN), by sonication for 24 h at 45 °C followed by ultracentrifugation for 8 h (Figure 1) [43].

### 2.3. Swelling Studies of Moringa oleifera Gum Nanogel (MOGN)

The gravimetric method was used to study the swelling behavior of MOGN [44]. For this, 0.01 g of the MOGN was immersed in distilled water, and after a specific time interval, the swollen MOGN was taken out from the solution and wiped with filter paper to remove excess water from its surface. Thereafter, the swollen nanogel was weighed on an electronic weighing balance (Explorer® Analytical and Precision, Ohaus Corporation, New York, NY, USA). The swollen nanogel was dipped into the distilled water again, and the process was repeated until MOGN attained a constant weight. The effect of pH on the swelling behavior of MOGN was investigated using a swelling medium of different pH (2.4, 3.5, 4.5, 5.0, 5.5, 6.8, and 7.4) and distilled water (pH 6.0) in order to mimic the physiological pHs, viz. the gastrointestinal tract, cancer cells, and interstitial fluid, with time at 37 °C. The swelling ratio (S_r_) and %swelling (P_s_) was calculated using the following equations [22,24,46]:(1)$${\;S}_{r}=\frac{w_{s}-w_{o}}{w_{o}}\;$$
(2)$${\;P}_{s}=\frac{w_{s}-w_{o}}{w_{o}}\times 100\;$$
where *w_s_* and *w_o_* are the weight of the swollen and dry nanogel, respectively.

### 2.4. DOX Loading and Release Studies

For the loading of the DOX onto MOGN, 50 mg of MOGN was immersed in 250 mg/L 50 mL DOX solution (prepared in 2% DMSO and phosphate buffer saline solution of pH 7.4) at room temperature. After specific time intervals, the absorbance of the solution containing the residual drug was measured on a UV-Vis spectrophotometer (Photolab 6600) at 495 nm. The amount of the unloaded drug was calculated from a calibration curve. After the optimum drug loading, the DOX-loaded MOGN, i.e., DOX–MOGN, was filtered and washed with distilled water to remove any free drug from its surface. Thereafter, DOX-MOGN was dried at room temperature. The drug loading capacity (*q*) and loading efficiency (%L) were determined from the expressions [24]:(3)$$q=\frac{C_{o}-C_{e}}{W}\times V\;$$
(4)$$\%L=\frac{C_{o}-C_{e}}{C_{o}}\times 100\;$$
where, C_o_ is the initial concentration of the drug loaded in time T = 0. C_e_ is the concentration of the drug remaining in the solution (mg/L) in time ‘t’. *V* is the volume of the aqueous phase, and *W* is the amount of dry MOGN.

DOX release studies were carried out in phosphate buffer solutions of pH 5.5, 6.8, and 7.4 to replicate pH values of an intracellular environment of compartments (endosomes and lysosomes), extracellular pH in tumor tissues and the physiological pH in normal tissues, respectively [16,27]. Different drug-loaded samples were separately studied for drug release at the physiological temperature (37 °C) for different time intervals. A known weight of the drug-loaded material was immersed in a specific pH solution, and the amount of released drug was estimated by measuring the absorbance after specific time intervals at 495 nm. %Release (P_r_) of the drug was calculated as [26]:(5)$${\;P}_{r}=\frac{C_{t}}{C_{o}}\times 100\;$$
where, C_t_ is the concentration of the drug released in time ‘t’. To understand the kinetics and release mechanism of DOX from DOX–MOGN, various kinetic models [47] viz. zero-order [48], first-order [49], Higuchi [50], and the Korsmeyer–Peppas model [51] were applied, and the respective equations are mentioned in Table 1. All the experiments studies were carried out in triplicate, and the results reported are the mean ± S.D.

### 2.5. Characterization Studies

Synthesized materials were characterized using different techniques to get evidence of MOGN synthesis. Fourier transform infrared (FTIR) spectra of the dry samples were recorded in the transmission mode on a FTIR Spectrophotometer Perkin Elmer Spectrum RX1 (Waltham, MA, USA), between 4000 and 500 cm^−1^ using the KBr pellets under a 300 kgf/cm^2^ hydraulic pressure. The surface morphologies of the samples (before and after DOX loading) were observed by field emission scanning electron microscopy (FESEM) images, mapping images, and energy–dispersive X-ray (EDS) spectra and were recorded with Hitachi SU8010 Series FESEM, Tokyo, Japan, at 15 kV. Thermal analysis of the samples was investigated with the TGA/SDTA 851 Mettler-Toledo thermal analyzer from room temperature to 600 °C at a heating rate of 10 °C/min in a stream of nitrogen (60 mL/min). The obtained TGA data were analyzed using the Mettler–Toledo Star System SW 15.00.

### 2.6. Antitumor Activity Study

#### 2.6.1. Cell Culture

RD cells were cultured in cell culture plates containing DMEM medium with 10% fetal bovine serum (FBS), 1% penicillin, and streptomycin. Thereafter, the cells were incubated for 24 h under a humidified atmosphere at 37 °C with 5% CO_2_ atmosphere [52].

#### 2.6.2. Anti–Tumor Activity Assay

The in vitro antitumor activity of free DOX and DOX–MOGN was carried out via MTT assay to determine the viability of RD cancer cell line in the presence of DOX–MOGN using MTT dye which on reduction by mitochondrial dehydrogenase present in living cells forms blue colored formazan crystals [52]. RD cells were suspended in a final concentration of 1.4 × 10^4^ cells/mL in DMEM. This cell suspension was seeded in 96–well plates (200 µL cell suspension/well). The cells seeded in the wells were allowed to grow for 24 h. After incubating, the cells were treated with free DOX and DOX–MOGN at various DOX concentrations (0.01–100 μg/mL) for 48 h. Thence, MTT (5 mg/mL in distilled water; 20 µL) was added to each well, followed by incubation at 37 °C in a CO_2_ incubator for another 2 h in the dark. Thereafter, the medium was completely removed from each well, and the precipitated intracellular formazan crystals formed in each well were dissolved by adding DMSO (100 µL). After gentle shaking, the absorbance of all the wells was measured at 570 nm with an automated plate reader (Thermo Scientific Multiskan EX Microplate reader, Waltham, MA, USA). Untreated cells were used as the control for 100% cell viability [53]. The treated groups of cells were compared with the control group in the absence of DOX. The half–inhibitory concentration (IC_50_) was calculated using GraphPad Prism 7 software. The growth inhibitory ratio, i.e., cell viability (%), was calculated using the following equation [24]:(6)$$\mathrm{Cell}\;\mathrm{viability}\;\left(\%\right)=\frac{A\;\left(\mathrm{Sample}\right)}{A\;\left(\mathrm{Control}\right)}\times 100\;$$
where *A* (Control) and *A* (Sample) are the absorbance values for the untreated cells and treated cells, respectively; these values were obtained after subtracting the absorbance value for DMSO.

The DOX amount calculated by Graph software based on the IC_50_ value was added to the cell culture in a tissue culture flask that contained DMEM low glucose medium (10 mL) marked as a test, and an equal volume of potassium phosphate buffer with PLP was added to the control tissue culture flask. The RD cells with free DOX and DOX–MOGN treatment were examined under the inverted microscope (Inverted Microscope, Hund Wilovert S, Wetzlar, Germany), and images were captured using a camera connected to a laptop with the help of the Software, Magnus Pro 3.0, Olympus. All the assay experiments were carried out in triplicate. 

## 3. Results and Discussion

### 3.1. Synthesis of Moringa oleifera Gum Nanogel (MOGN)

Superabsorbent MOGN was synthesized using the γ–irradiation initiation method that involves a free radical mechanism. γ–rays generate free radicals, viz. hydroxyl radical from water and alkoxy radical from MOG, by extracting hydrogen from the –OH groups present in its skeleton. The hydroxyl radicals also abstract hydrogen from the MOG and form an alkoxy radical. These free radicals then attack the vinyl group of acrylamide (CH_2_=CHCONH_2_, AAm) to generate a new radical on the monomer surface that initiates polymerization. Simultaneously, N,N’-MBA reacts with the polymer chain via its vinyl groups by linking with the poly(AAm) and MOG to generate a three-dimensional polymeric network, MOGH. On hydrolysis with NaOH, the –NH_2_ groups of MOGH were partially converted into ionized –COOH groups [45]. The hydrolyzed MOGH was converted into nanogel, MOGN, by subjecting it to sonication followed by ultracentrifugation (Figure 2).

### 3.2. All Characterization Studies

#### 3.2.1. FTIR Analysis

FTIR spectrum of MOGH has characteristic bands at 3334 cm^–1^ (–N–H stretching vibrations of primary amide), 1657 cm^–1^ (–C=O stretching vibrations of amide II band), 1603 cm^–1^ (N–H bending vibrations), 1421 cm^–1^ (N–H bending vibrations of amide III), 1311 cm^–1^ (C–N stretching vibrations of secondary amide due to the presence of N,N’-MBA) which were absent in pure MOG, along with the characteristic bands of pristine MOG at 1451 cm^–1^ (–COO^–^ stretching vibrations due to uronic acid of MOG), 1040 cm^–1^ (complex band due to C–O and C–O–C stretching vibrations), and 885 cm^–1^ (pyranose ring modes of polysaccharide skeleton) [45,54]. Whereas, in the FTIR spectrum of MOGN disappearance of the band at 3334 cm^–1^ and 1603 cm^–1^, along with the appearance of a new band at 1553 cm^–1^ (–COO^–^ antisymmetric vibrations) and 1405 cm^–1^ (–COO^–^ symmetric vibrations of ionized –COOH groups) confirms the partial hydrolysis of –NH_2_ groups of poly(Aam) [45]. The FTIR spectrum of DOX–MOGN has all the absorption bands of MOGN with a slight change in their position and intensity (Figure 1). Thus, FTIR spectral studies confirm the successful synthesis of MOGN and the loading of DOX on it.

#### 3.2.2. SEM Analysis

From the SEM image of MOGN, it can be seen that particles are spherical in shape with a size of ˂45 nm (Figure 2a), which confirms the nano–dimensional spherical structure of the synthesized nanogel, thus indicating its suitability for drug delivery applications. While the SEM image of DOX–MOGN showed a somewhat rough surface due to the deposition of the drug molecules on its surface (Figure 2b). In addition, the successful loading of DOX onto MOGN was further confirmed by EDS spectra of MOGN and DOX–MOGN.

#### 3.2.3. EDS and Elemental Mapping Analysis

The EDS spectrum of MOGN displayed a new peak of N due to the incorporation of acrylamide and N,N’-MBA along with the characteristic peaks of C and O in the EDS spectrum of MOG [54], indicating its successful synthesis from MOG. Furthermore, a change in the %weight of C, O, and N in the EDS spectrum of DOX–MOGN confirmed the successful loading of DOX onto MOGN (Figure 3a,b). Elemental mapping of MOGN and DOX–MOGN validates the presence of C, O, and N elements and increases the uniform distribution of N in DOX–MOGN than MOGN, confirming the effective loading of DOX onto MOGN (Figure 4a,b).

#### 3.2.4. Thermal Analysis

Thermal analysis of the pure and hydrolyzed grafted form of MOG was investigated using TGA and DTG curves and is depicted in Figure 5a,b. Based on the TGA and DTG curves for pure MOG, two mass loss events were observed wherein the first initial mass loss of 10.9% occurred at >150 °C with a peak at the maximum decomposition temperature (T*_max_*) of 65.99 °C due to the evaporation of moisture content and the second major mass loss occurred between 253 °C–400 °C (final decomposition temperature, T*_f_*) are associated with the decomposition of the polysaccharide backbone of MOG. After that, the residual part degraded slowly up to 550 °C, leaving a constant residual mass of 24.4% [55]. In addition, in MOG, a 50% mass loss was observed at 309.68 °C, and this temperature is designated as T_50_. In the thermogram of the hydrolyzed grafted form, i.e., MOGN, additional multiple mass loss steps were observed. The initial mass loss of 5% due to evaporation of water occurred at >100 °C with T*_max_* at 66.21 °C, and second and third mass losses occurred from 140 °C to 200 °C and from 256.74 °C to 392.71 °C with T*_max_* at 156 °C and 301.15 °C. They are associated with the degradation of the N,N’-MBA and polyacrylamide chains in the MOGN skeleton, respectively. The fourth mass loss up to 500 °C (final decomposition temperature, T*_f_*) is attributed to the degradation of the remaining polymer backbone. Thereafter, the slow degradation of residual mass was continued till 600 °C [23,56]. In addition, in the thermogram of MOGN, 50% mass loss occurred at 443.32 °C (T_50_), which was higher than pure MOG. Hence, the T_50_ and T*_f_* were higher for MOGN than its pure form, thus, indicating that incorporation of polyacrylamide and N,N’-MBA enhanced the thermal stability of the synthesized MOGN.

### 3.3. Swelling Studies of MOGN

Synthesized MOGN behaves as a superabsorbent nanogel as it can hold a huge amount of water in its network with a maximum uptake of 365.0 g/g (Figure 6a,b). Such a high superabsorbent property of water uptake comes from the partial hydrolysis of the –NH_2_ groups on the polymer skeleton, resulting in the generation of COO^–^ groups. The ionized –COOH groups thus formed, and the innate carboxylic groups of glucuronic acid in the polymer skeleton of MOG bestowed a huge water-absorbing capacity to MOGN [57]. The equilibrium P_s_ of 650; 910; 2100; 5370 at pH 2.4, 3.5, 4.5, 5.0 in 1080 min (18 h) and 10,500; 36,300; 9700 and 4250 at pH 5.5, distilled water (6.0), 6.8, 7.4 was attained in 4320 min (72 h) at 37 °C, respectively (Figure 6c,d). From the results, it is revealed that the maximum P_s_ of MOGN at 37 °C with pH-responsiveness to water uptake was in order: 6.0 > 5.5 > 6.8 > 7.4 > 5.0 > 4.5 > 3.5 > 2.4. Such a trend of swelling of the MOGN can be attributed to its changing behavior with pH. At low pH, the suppression of its ionized carboxylic groups occurs due to the presence of H+ ions in the external medium. At an alkaline pH, the interactions of the unionized carboxylic groups with water molecules are hindered due to the presence of basic moieties on the surface of nanogel. Thus, the overall effect results in the highest swelling at pH 6.0 [46]. Thus, the water absorption capacity of MOGN is pH-dependent, indicative of its stimuli–responsive swelling nature. These features, viz. high-water adsorption capacity and pH-responsive nature of the synthesized material in the pH range of cancerous cells, 5.0–6.8, are the two important attributes that make it the proficient candidate for the site-specific drug delivery platform to ensure the effective release of DOX.

### 3.4. DOX Loading and Release Studies

DOX loading onto MOGN and its release from there have been schematically presented in Figure 3. This is attributed to the presence of sufficiently available ionized carboxylic groups (–COO^−^) present in the MOGN skeleton, which attracts the protonated amino groups (–NH_3_^+^) of DOX via electrostatic interactions [24,58]. It was observed that DOX-MOGN showed a very high %loading (%L) of 98.35%of DOX on MOGN (Figure 7a). UV–spectral studies further confirmed that the DOX loading onto MOGN was observed from a decrease in the absorbance of the DOX concentration with time (Figure 7b). Pikabea et al. observed a similar trend for loading efficiency for PDEAEMA-based nanogel; however, the synthesized nanogel, MOGN, showed a higher %L than PDEAEMA-based nanogel (90%) and other reported nanogels in the literature elsewhere [22,23,28,30]. Thus, the MOGN fulfills the primary objective of high drug loading for effective dosage for the localized release at the affected part of the body.

To evaluate the pH-triggered DOX release behaviour, in vitro release studies of DOX from DOX-MOGN with time was investigated at 37 °C under different pH media of 5.5, 6.8, and 7.4 [16,27]. From the release profile, it was observed that the release of DOX from DOX–MOGN was both time and pH–dependent. About 9.06% in 24 h and 12.18% in 72 h release of DOX from MOGN was observed at pH 7.4. This can be ascribed to the equilibrium partitioning effect between NH_3_^+^ groups of DOX and COO^-^ groups of the nanogel. At pH 5.5 and pH 6.8, DOX release was found to be much higher, i.e., 83.42% and 53.72% in 24 h, and 91.92% and 62.62% in 72 h, respectively, due to the combined effect of the weakening of electrostatic interactions between the DOX and nanogel at the acidic pH along with the pH-triggered swelling-cum-disintegration of the MOGN (Figure 8a) [25,58]. Therefore, it can be concluded that the amount of DOX released was maximum at pH 5.5 and least at pH 7.4. This trend is attributed to the combined effect of enhanced swelling at acidic pH, diffusion of the drug, and nanogel disintegration [27]. Since DOX release was maximum at pH 5.5, the stability of this pH-responsive system at pH 7.4 was advantageous as DOX was not released during blood circulation and would lessen the toxic effects associated with the free drug.

Plots of different drug-release kinetic models are shown in Figure 8b–e. Therefrom, it can be observed that at pH 5.5 and 6.8, the release of DOX from DOX–MOGN followed the Korsmeyer–Peppas release model with the highest value of correlation coefficient, R^2^, 0.99, and 0.952, respectively (Table 2). The best fit of the Korsmeyer–Peppas model indicated the diffusion and the swelling-controlled release of DOX from DOX–MOGN, further validated by the values of n obtained from the slope of the Korsmeyer–Peppas plot. From Table 2, the values of n at pH 5.5 and 6.8 were 0.61 and 0.75, respectively, suggesting an anomalous non–Fickian transport mechanism, meaning that the drug release mechanism was governed by diffusion and swelling of DOX–MOGN [59,60]. But at pH 7.4, the release of DOX from DOX–MOGN followed the Higuchi kinetic model with the highest R^2^ value of 0.956, suggesting a diffusion-controlled release mechanism [61].

A comparison of %L and P_r_ of DOX for various nanogels with the synthesized MOGN was presented in Table 3. Therefrom, it was revealed that the synthesized MOGN is an efficient DOX delivery device with a high loading efficiency of 98.35% and P_r_ of 91.92%, 62.62%, and 12.18% at pH 5.5, 6.8, and 7.4 after 72 h, respectively.

### 3.5. In Vitro Cytotoxicity Studies

The cytotoxicity of the DOX delivery platform was investigated and compared with the free DOX using RD cells. Normal RD cells had a spindle shape, but after treatment with free-DOX and DOX-MOGN, the cell shape got distorted as the dead cells assumed a spherical shape (Figure 9a–c). The cytotoxicity of DOX-MOGN was concentration-dependent, and cell viability of 7.0% at pH 5.5 was observed for DOX-MOGN (Figure 9d) [70]. Moreover, IC_50_ of RD cells treated with DOX-MOGN (0.946 μg/mL) at pH 5.5 was near to the IC_50_ of free DOX (0.812 μg/mL), confirming the maximum release of DOX from MOGN at pH 5.5 [16]. Thus, in vitro cytotoxicity studies displayed close consonance with in vitro release studies. Hence, the effective release at intracellular pH 5.5 makes MOGN an excellent DOX nanocarrier for its pH-triggered release at the cancer site.

## 4. Conclusions

In the present research work, *M. oleifera* gum nanogel, MOGN, was successfully synthesized via γ–irradiation, followed by partial hydrolysis, sonication, and ultracentrifugation. Thereafter, MOGN was evaluated as a nanocarrier for DOX delivery. MOGN showed a high swelling ratio within the pH range of cancerous cells, i.e., 5.0–6.8, with a maximum swelling ratio of 365.0 g/g at pH 6.0 and 37 °C. MOGN exhibited a very high %L (98.35%) of DOX. The release kinetics of DOX from DOX–MOGN at pH 5.5 and 6.8 follows the Korsmeyer–Peppas model, whereas at pH 7.4, it follows the Higuchi kinetic model. The DOX release from DOX–MOGN was the highest at simulated endosomal and extracellular pH of the tumor tissue, i.e., pH 5.5 (91.92%) and the lowest at pH 7.4 (12.18%), thereby indicating that DOX–MOGN can deliver DOX specifically to the tumor cells. In addition, the optimal DOX release and cytotoxicity studies at pH 5.5 are in consonance. The results obtained suggest that the pH-responsive *M. oleifera* nanogel is an effective intracellular drug delivery system for DOX cancer therapy and is capable of reducing the side effects of anticancer drugs.

## Data Availability

Not applicable.

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
