# Peer review of "Functionalized Moringa oleifera Gum as pH-Responsive Nanogel for Doxorubicin Delivery: Synthesis, Kinetic Modelling and In Vitro Cytotoxicity Study"

_polymers, 2022, doi:10.3390/polym14214697_

Round 1
Reviewer 1 Report
- There are some typo errors.
- Abstract conclusion is missing.
- Please use the full names of abbreviations before using abbreviations.
- free DOX and DOX–MOGN should be tested against normal cells.
- Why MOGN did not enhance the cytotoxicity of DOX in RD cells.
- The rationale for using RD cells is missing, why not the other kind of cells?
Reviewer 2 Report
Although several research has been published on Moringa oleifera gum grafted with acrylamide using free radical grafting techniques, the results presented by authors are interesting.
The authors failed to bring the novelty of the current work into the abstract and introduction sections. Suggested improving both sections.
Moreover, in the method section, it is suggested to elaborate on the characterization sections in a concise manner.
In addition, suggested adding NMR analysis to confirm the grafting
Further, suggested performing a thermal analysis with a comparison between native and graft forms.
Furthermore, an improvement in the discussion section requires, comparing the results with the previously published paper does not reflect exactly how the previous work significantly differs. The significant difference requires a brief discussion to prove the concept of current work worth.
Some grammatical errors have been reflected in the attached PDF version. Suggested to consider while doing revisions

Round 2
Reviewer 1 Report
The authors addressed all the comments.
Reviewer 2 Report
Authors have reflected required corrections / suggestions. The manuscript can be accepted in present form.